# Liquid biopsy tracking during sequential chemo-radiotherapy identifies distinct prognostic phenotypes in nasopharyngeal carcinoma

Jiawei Lv[1,3], Yupei Chen[1,3], Guanqun Zhou[1,3], Zhenyu Qi[1,3], Kuan Rui Lloyd Tan [2], Haitao Wang [2], Li Lin[1], Foping Chen[1], Lulu Zhang[1], Xiaodan Huang[1], Ruiqi Liu[1], Sisi Xu[1], Yue Chen[1], Jun Ma[1], Melvin L.K. Chua [2] & Ying Sun [1]

Liquid biopsies have the utility for detecting minimal residual disease in several cancer types. Here, we investigate if liquid biopsy tracking on-treatment informs on tumour phenotypes by longitudinally quantifying circulating Epstein-barr virus (EBV) DNA copy number in 673 nasopharyngeal carcinoma patients undergoing radical induction chemotherapy (IC) and chemo-radiotherapy (CRT). We observe significant inter-patient heterogeneity in viral copy number clearance that is classifiable into eight distinct patterns based on clearance kinetics and bounce occurrence, including a substantial proportion of complete responders (≈30%) to only one IC cycle. Using a supervised statistical clustering of disease relapse risks, we further bin these eight subgroups into four prognostic phenotypes (early responders, intermediate responders, late responders, and treatment resistant) that are correlated with efficacy of chemotherapy intensity. Taken together, we show that real-time monitoring of liquid biopsy response adds prognostic information, and has the potential utility for risk-adapted treatment de-intensification/intensification in nasopharyngeal carcinoma.

[1] Department of Radiation Oncology, Sun Yat-sen University Cancer Center; State Key Laboratory of Oncology in South China, Collaborative Innovation Center for Cancer Medicine, Guangdong Key Laboratory of Nasopharyngeal Carcinoma Diagnosis and Therapy, Guangzhou, People's Republic of China. [2] Division of Radiation Oncology, National Cancer Centre Singapore; Division of Medical Sciences, National Cancer Centre Singapore, Duke-NUS Medical School, 11 Hospital Drive, Singapore 169857, Singapore. [3]These authors contributed equally: Jiawei Lv, Yupei Chen, Guanqun Zhou, Zhengyu Qi. Correspondence and requests for materials should be addressed to M.L.K.C. (email: melvin.chua.l.k@singhealth.com.sg) or to Y.S. (email: sunying@sysucc.org.cn)

Endemic nasopharyngeal carcinoma (NPC) is invariably associated with Epstein–Barr virus (EBV) infection. Circulating cell-free EBV DNA (cfEBV DNA) consisting of short DNA fragments released by NPC cells can be detected using ultrasensitive polymerase chain reaction (PCR)-based assays[1,2]. Detection of this biomarker has demonstrated clinical utility for population screening[3], risk stratification[4,5] and disease surveillance[1,6,7]. In particular, detectable cfEBV DNA following chemo-radiotherapy (concurrent chemo-radiotherapy (CCRT)) is thought to suggest minimal residual disease[7–10]. This prompted NRG to conduct a clinical trial (NRG-HN-001, NCT02135042, ClinicalTrials.gov) investigating the role of this biomarker for systemic chemotherapy de-intensification (undetectable post-CCRT cfEBV DNA) and intensification (detectable post-CCRT cfEBV DNA). Of note, Chan and colleagues reported on the results of their multicentre randomised controlled trial (NPC-0502), indicating no difference in clinical outcomes with additional gemcitabine and cisplatin chemotherapy compared to observation for high-risk patients with minimal residual disease following radiotherapy[11]. Despite the lack of efficacy, caveats of the risk-stratification approach employed in the trial included the extremely low rates of accrual (only 104 of the 798 screened patients enrolled) and poor tolerability to adjuvant chemotherapy (AC; only 50% received all 6 cycles) following an intensive course of CCRT. These reasons support the rationale to investigate whether cfEBV DNA can be employed earlier for risk stratification and treatment adaptation.

With the emergence of clinical evidence supporting the efficacy of induction chemotherapy (IC) and CCRT in locally advanced NPC (LA-NPC)[12–15], we therefore investigate whether cfEBV DNA response to IC harboured prognostic significance. First, we characterise the longitudinal changes of this circulating tumour marker to IC and CCRT in 673 patients identified from a single academic institution Big-data platform and subsequently evaluate the association of differential cfEBV DNA responses with risks of relapse. We define four distinct phenotypic clusters of patients based on their onset of complete biological response (cBR; defined as undetectable cfEBV DNA) to treatment and further demonstrate the potential of our risk grouping for stratification to treatment adaptation in LA-NPC patients.

## Results

**Patient and treatment characteristics.** We identified 673 eligible patients from 10,126 non-metastatic NPC patients screened from an NPC-specific database within the Big-data intelligence framework; the stepwise selection process are presented in Supplementary Fig. 1; the collection schema of cfEBV DNA is presented in Supplementary Fig. 2. Baseline characteristics of the 673 LA-NPC cases are summarised in Supplementary Table 1; it comprised of 575 (85.4%) patients with T3–4 tumours and 402 (59.7%) patients with N2–3 tumours. Median pretreatment cfEBV DNA was $12.9 \times 10^3$ copies/ml (range: $0.05 \times 10^3$–$9080 \times 10^3$ copies/ml), and was correlated to Tumour Node Metastasis stage (median cfEBV DNA $= 9.24 \times 10^3$, stage III vs. $20.25 \times 10^3$, stage IV$_A$, $P = 0.01$, Mann–Whitney $U$ test). Median follow-up duration was 42.4 months (interquartile range 28.6–52.6 months). We recorded 86 locoregional recurrences, 112 distant metastases and 25 synchronous locoregional and metastatic recurrences. Three-year disease-free survival (DFS), overall survival (OS), distant metastasis-free survival (DMFS) and locoregional relapse-free survival (LRFS) for the cohort were 76.2, 88.7, 84.6 and 89.1%, respectively.

In terms of treatment characteristics, more than half received triplet docetaxel–cisplatin–fluorouracil (TPF) IC ($N = 387$, 57.5%); for doublet IC regimes, these included docetaxel–cisplatin (TP) ($N = 176$, 26.2%), cisplatin–fluorouracil (PF) ($N = 58$, 8.6%) and gemcitabine–cisplatin (GP) ($N = 45$, 6.7%) in descending order. For the number of IC cycles, 353 (52.5%) patients received two cycles of IC, 268 (39.8%) received three cycles of IC and 52 (7.7%) received four cycles of IC (Supplementary Table 1). Dose modifications during the IC phase were required for 83 (12.3%) patients (Supplementary Fig. 3). Cisplatin was the commonest drug of choice for combination with radiotherapy ($N = 591$, 87.8%; [$N = 496$, 3-weekly; $N = 95$, weekly]); nedaplatin ($N = 82$, 12.2%; [$N = 74$, 3-weekly; $N = 8$, weekly]) was the other concurrent systemic agent. Majority of cohort received ≥160 mg/m² cumulative concurrent dose (CCD) of cisplatin/nedaplatin ($N = 504$ [74.9%]; $N = 169$ [25.1%] CCD < 160 mg/m²; Supplementary Fig. 3).

**Longitudinal cfEBV DNA defines distinct response phenotypes.** Figure 1 illustrates the longitudinal cfEBV DNA trend over the IC and CCRT treatment phases. Four hundred and twenty-five of the 673 patients (63.2%) achieved cBR in the IC phase; 245 patients (of 425, 57.6%) by IC$_1$ and 139 patients (32.7%) by IC$_2$. Interestingly, we observed a bounce phenomenon (a positive cfEBV DNA reading following cBR) in 39 of 425 patients with cBR during the IC phase (Fig. 1 and Supplementary Fig. 4); nonetheless, almost all bounces ($N = 37$) resolved post-CCRT. For patients with persistent cfEBV DNA detectable titres post-IC$_2$ ($N = 307$; includes 18 patients with bounce after cBR at IC$_1$), 160 (52.1%) patients received a further 3–4 cycles of IC, while 147 (47.9%) patients proceeded to undergo CCRT; of these, 48 achieved cBR post-IC$_{3-4}$, and 118 achieved cBR post-CCRT.

Of the 248 patients with detectable cfEBV DNA titres prior to CCRT, 193 (77.8%) achieved cBR post-CCRT. A cfEBV DNA bounce was also observed in 42 patients during the CCRT phase (delayed bounce). In contrast to patients with early bounce during IC, only 22 patients had a subsequent cBR for delayed bounces (Fig. 1). Finally, a minority of patients ($N = 44$, 6.8%) reported persistent cfEBV DNA post-IC+CCRT. Based on these observations, we defined eight distinct cfEBV DNA response phenotypes: Group 1 (G1, $N = 200$; 29.7%), cBR post-IC$_1$ without bounce; G2 ($N = 113$; 16.8%), cBR post-IC$_2$ without bounce; G3 ($N = 43$; 6.4%), cBR post-IC$_{3-4}$ without bounce; G4 ($N = 117$; 17.4%), cBR post-CCRT+IC$_2$; G5 ($N = 75$; 11.1%), cBR post-CCRT+IC$_{3-4}$; G6 ($N = 59$; 8.8%), temporary bounce with cBR post-CCRT; G7 ($N = 22$; 3.3%), persistent bounce with non-cBR post-CCRT; G8 ($N = 44$; 6.5%), persistent DNA, defined as detectable cfEBV DNA (>0 copies/ml) throughout and following the course of IC and CCRT (Fig. 1).

Next, we determined the prognostic significance of detectable cfEBV DNA at the respective time points. In keeping with previous observations, pretreatment cfEBV DNA stratified by 2000 copies (>2000 vs. ≤2000 copies/ml) was significantly associated with DFS (adjusted hazard ratio (AHR) = 2.06, 95% confidence interval [CI] = 1.32–3.54, $P < 0.01$, corrected for age, sex and T and N categories; Table 1). In addition, we observed that cBR throughout the different phases of treatment (post-IC$_1$, post-IC$_2$, post-IC$_{3-4}$, pre-RT and post-RT) conferred a favourable prognosis compared to non-responders ($P_{all} < 0.01$; Fig. 2). Persistent post-CCRT cfEBV DNA was the most adverse prognostic factor across all survival endpoints (AHR$_{DFS} = 5.30$, 95% CI = 3.78–7.44, $P < 0.01$; Table 1).

**cfEBV DNA response phenotypes predict differential prognoses.** Next, we investigated the prognostic association of the eight biological response phenotypes with DFS in our cohort. We observed that prognoses differed significantly between them (Fig. 3a). In particular, patients with cBR post-IC$_1$ (G1) were

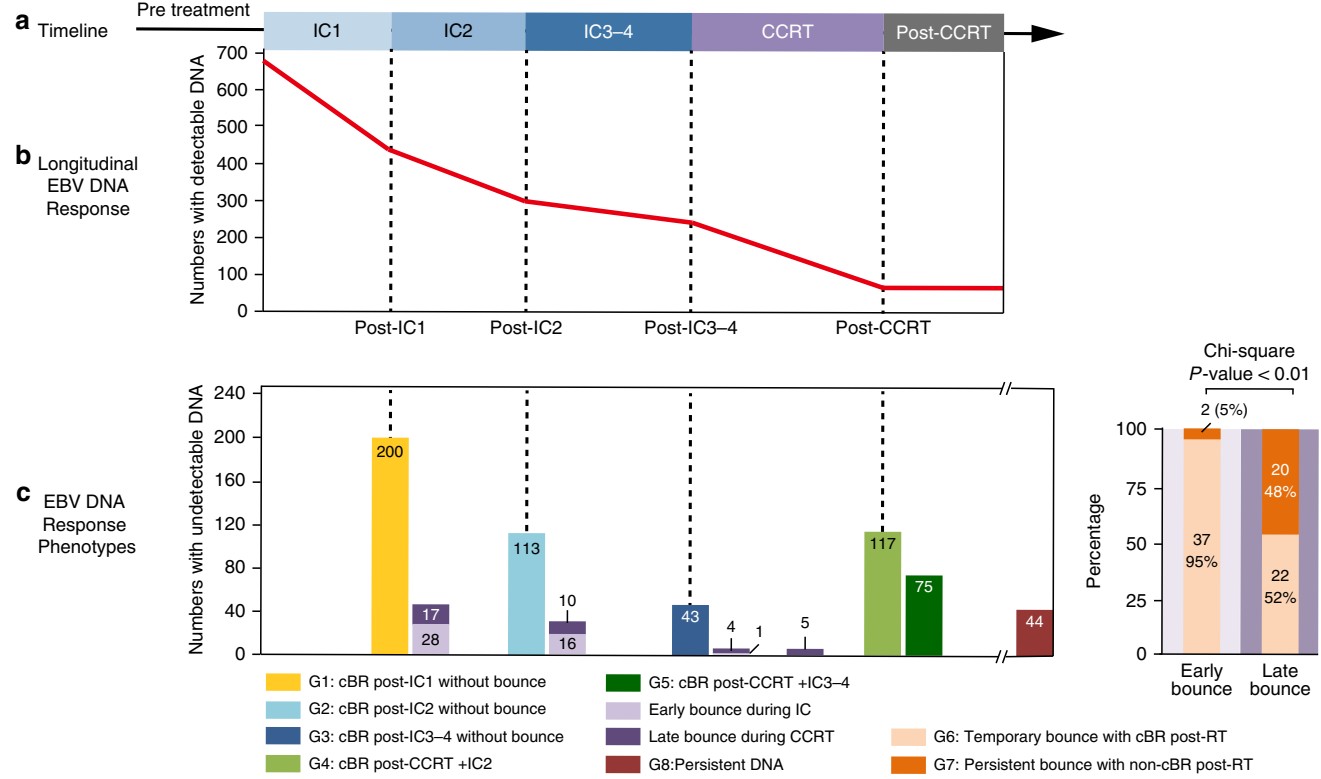

**Fig. 1** Longitudinal map for onset of complete biological response in nasopharyngeal carcinoma patients. **a** Treatment timeline. **b** The longitudinal cell-free Epstein–Barr virus (cfEBV) DNA responses during induction chemotherapy and concurrent chemo-radiotherapy. **c** Differential cfEBV DNA response phenotypes characterised by inter-individual variation in complete biological response (cBR) and onset of bounce (a positive cfEBV DNA reading following cBR) (left panel), and characterisation of two types of EBV DNA bounce by biological response post radiotherapy (right panel). Six hundred and seventy-three locally advanced nasopharyngeal carcinoma patients receiving induction chemotherapy and concurrent chemotherapy were included. cBR was defined by undetectable cfEBV DNA level

most favourable among all, even when compared to patients from G2 ($HR_{DFS} = 0.37$, 95% CI = 0.19–0.74, $P < 0.01$). Interestingly, patients from G5 (cBR post-CCRT+$IC_{3-4}$) had a higher risk of relapse compared to those from G4 (cBR post-CCRT+$IC_2$) ($HR_{DFS} = 2.18$, 95% CI = 1.34–3.54, $P < 0.01$), which may be attributed to chemotherapy-resistant tumour clones (persistent cfEBV DNA despite more IC cycles) and/or accelerated tumour repopulation[16]. Among patients demonstrating a bounce phenomenon, G6 had significantly better survival compared to G7 ($HR_{DFS} = 0.38$ for temporary vs. persistent, 95% CI = 0.20–0.72, $P < 0.01$). On this note, we observed that G7 comprised of predominantly delayed bounces during CCRT, and patients in this subgroup did not harbour unusually high cfEBV DNA at baseline and in fact demonstrated comparable response to $IC_1$ against the other subgroups (Fig. 1, Table 2). Collectively, this puts forth the concept that delayed bounces could be indicative of treatment-refractory tumour clones. Consistent with this notion, comparable outcomes were observed between G7 and G8 patients.

Given the overlap in DFS between the phenotypes (Fig. 3a and Supplementary Fig. 5), we performed a clustering analysis based on the relative $HR_{DFS}$ between each phenotype to reduce the number of subgroups (Fig. 3b). We obtained four distinct clusters comprising of the following phenotypes: G1 as Cluster 1; Cluster 2 was comprised of G2, G3, G4 and G6; G5 as Cluster 3; and Cluster 4 consisted of G7 and G8. The four clusters demonstrated significant different inter-group prognoses across all clinical endpoints (3-year DFS = 94.3, 78.0, 60.5, 25.1%, respectively, $P < 0.01$ by landmark analysis; Fig. 3c and Supplementary Fig. 5). Coincidentally, these four distinct clusters were closely correlated

with clinical observations. Therefore, we termed them as such: Cluster 1 as early responders ($N = 200$, 29.7%); Cluster 2 as intermediate responders ($N = 332$, 49.3%); Cluster 3 as late responders ($N = 75$, 11.1%); and Cluster 4 as treatment-resistant ($N = 66$, 9.8%).

Finally, we tested for potential associations between the respective cfEBV DNA response phenotypes and clinical covariates that may influence the tumour marker response, including pretreatment cfEBV DNA load, T and N categories, IC and concurrent chemotherapy intensity (doublet vs. triplet, modified vs. non-modified IC dose, CCD < 160 mg/m² vs. ≥160 mg/m²). Unsurprisingly, pretreatment cfEBV DNA and N status were associated with the four clusters (Fig. 3e and Table 2); in particular, a higher baseline cfEBV DNA load corresponded to a lower likelihood of being an early responder and a higher likelihood of being treatment-resistant ($P < 0.01$, Chi-squared test; Fig. 3d). Detailed information of baseline pretreatment cfEBV DNA loads, the absolute drop of cfEBV DNA during the course of treatment and T and N categories among the eight response groups/four phenotypic clusters is presented in Table 2 and Supplementary Table 2. Chemotherapy intensity, treatment interruption and prolonged wait time to radiotherapy were not associated with the response phenotypes ($P > 0.05$; Fig. 3f and Supplementary Tables 3 and 4); and separately, we did not observe an association between achieving a complete biological response post-$IC_1$ and post-$IC_2$ and the number of IC cycles received ($P_{both} > 0.05$; Supplementary Table 5). Our four phenotype clusters remained significantly associated with prognoses, adjusted for pretreatment cfEBV DNA load, chemotherapy intensity and T and N categories on multivariable analyses

**Table 1 Cox proportional hazard analyses of the longitudinal cfEBV DNA response in 673 locoregionally advanced nasopharyngeal carcinoma patients**

| Longitudinal cfEBV DNA | $AHR_{DFS}$ (95% CI) (events, $N = 178$) | P value | $AHR_{OS}$ (95% CI) (events, $N = 96$) | P value | $AHR_{DMFS}$ (95% CI) (events, $N = 112$) | P value | $AHR_{LRFS}$ (95% CI) (events, $N = 86$) | P value |
|---|---|---|---|---|---|---|---|---|
| *DNA responses at different time points* | | | | | | | | |
| Pretreatment (>2000 vs. ≤2000 copies/ml)[a] | 2.06 (1.32–3.54) | <0.01 | 1.99 (1.03–3.99) | 0.04 | 2.46 (1.57–9.37) | <0.01 | 1.60 (0.86–2.98) | 0.14 |
| Post-$IC_1$ (cBR vs. non-cBR)[b] | 2.71 (1.82–4.03) | <0.01 | 3.13 (1.73–5.66) | <0.01 | 2.56 (1.66–4.60) | <0.01 | 2.73 (1.61–4.98) | <0.01 |
| Post-$IC_2$ (cBR vs. non-cBR)[b] | 2.69 (1.95–3.72) | <0.01 | 2.88 (1.83–4.52) | <0.01 | 2.62 (1.74–3.87) | <0.01 | 2.55 (1.84–4.74) | <0.01 |
| Post-$IC_{3-4}$ (cBR vs. non-cBR)[b] | 3.93 (2.57–6.01) | <0.01 | 4.68 (2.62–8.40) | <0.01 | 4.22 (2.54–7.03) | <0.01 | 3.89 (2.25–8.18) | <0.01 |
| Pre-RT (cBR vs. non-cBR)[b] | 2.74 (1.88–3.45) | <0.01 | 2.90 (1.90–4.43) | <0.01 | 2.68 (1.83–3.92) | <0.01 | 2.60 (1.86–4.53) | <0.01 |
| Post-RT (cBR vs. non-cBR)[b] | 5.30 (3.78–7.44) | <0.01 | 6.27 (4.08–9.65) | <0.01 | 6.65 (4.45–9.31) | <0.01 | 3.44 (2.02–5.85) | <0.01 |
| *DNA response phenotypes*[b] | | | | | | | | |
| Classification 1 (early responder) | Reference | <0.01 | Reference | <0.01 | Reference | <0.01 | Reference | <0.01 |
| Classification 2 (intermediate responder) | 3.46 (2.01–6.25) | | 5.52 (1.97–15.49) | | 3.05 (1.43–6.52) | | 3.43 (1.61–7.32) | |
| Classification 3 (late responder) | 7.50 (4.24–14.77) | | 11.44 (3.82–28.27) | | 7.76 (3.42–15.57) | | 6.28 (2.65–14.91) | |
| Classification 4 (treatment resistant) | 17.33 (10.06–33.38) | | 32.08 (11.22–65.72) | | 19.85 (9.12–42.23) | | 10.51 (4.49–24.90) | |

*AHR* adjusted hazard ratio, *CI* confidence interval, *cBR* complete biological response, *IC* induction chemotherapy, *DFS* disease-free survival, *DMFS* distant metastasis-free survival, *LRFS* locoregional relapse-free survival, *cfEBV DNA* cell-free Epstein–Barr virus deoxyribonucleic acid, *OS* overall survival
[a]Age (≥45 vs. <45 years), sex (male vs. female), T category (T4 vs. T3 vs. T1–2) and N category (N3 vs. N2 vs. N0–1) were included in the Cox regression model
[b]Pretreatment cfEBV DNA (>2000 vs. ≤2000 copies/ml), T category (T4 vs. T3 vs. T1–2), N category (N3 vs. N2 vs. N0–1), IC regimens (triplets vs. doublets) and cumulative concurrent chemotherapy dose (≥160 mg/m$^2$ vs. <160 mg/m$^2$) were included in the Cox regression model

($P < 0.01$; Table 1); other parameters associated with clinical outcomes are presented in Supplementary Table 6.

**Treatment adaption based on cfEBV DNA response clusters.** Next, we examined the relationship between the different cfEBV DNA responders and treatment intensity. Given the difference in disease relapse rates between intermediate (Cluster 2) and late responders (Cluster 3), we sought to interrogate the optimal chemotherapy intensity, as additional 3–4 cycles of IC were associated with inferior survival in G5 than in G4 patients (Fig. 3a). To this end, we compared the efficacy of 2 against 3–4 IC cycles (without IC regimens alteration) for patients with and without cBR post-$IC_2$. We observed no treatment differences in patients with cBR post-$IC_2$. (Fig 4a and Supplementary Table 7). However, patients with persistent tumour marker had inferior DFS despite more chemotherapy that was attributed to inferior distant metastasis control but not local relapse ($HR_{DFS} = 1.83$, $P < 0.01$; $HR_{DMFS} = 2.50$, $P < 0.01$; Fig. 4b and Supplementary Table 7). Next, for treatment-resistant patients (Cluster 4), additional AC to IC+CCRT improved distant metastatic control ($HR_{DMFS} = 0.42$, 95% CI = 0.24–0.74, $P < 0.01$; Fig. 4c).

Based on these exploratory observations, we propose a risk-stratified treatment adaptation design that is based on our phenotypic clusters and longitudinal surveillance of cfEBV DNA (Fig. 5). For early responders (Cluster 1), de-intensification may be considered given the superior survival of this favourable subgroup. Intermediate responders (Cluster 2) could represent a subgroup with high volume of occult tumour burden resulting in delayed cfEBV DNA clearance, and thus maintaining chemotherapy intensity with CCRT could be the optimal strategy. Chemotherapy resistance and/or accelerated repopulation are likely to account for Clusters 3 and 4, and therefore alternative systemic combinations (e.g. immunotherapy) with radiotherapy rather than additional IC cycles ought to be considered. Lastly, there may be a role for further AC intensification for the unfavourable Cluster 4.

## Discussion

Here we comprehensively profiled the longitudinal cfEBV DNA responses to IC and CCRT in 673 LA-NPC cases and observed eight distinct trajectories of biological responses to chemotherapy and radiotherapy that were characterised by inter-individual variation in clearance kinetics and onset of tumour marker bounce. These eight subgroups clustered into four cfEBV DNA response phenotypes (early, intermediate and late responders and treatment-resistant), defined primarily by their respective sensitivity to treatment. Importantly, the phenotypic clusters were associated with disparate risks of tumour relapse, which were dominated by distant metastatic recurrences. Consistent with these observations, the response phenotypes also corresponded to the efficacy of chemotherapy intensity for distant metastasis control; in particular, our exploratory analyses suggest that more of the same chemotherapy in non-cBR responders after two cycles of IC may not be effective in improving prognoses, and further systemic therapy intensification may be beneficial for the minority (<10%) of treatment-resistant patients. Taken together, our findings timely demonstrated the feasibility and clinical impact of longitudinal liquid biopsies that can inform on outcomes following treatment. In addition, the utility of real-time information on treatment response for both treatment intensification and de-intensification ought to be investigated in a multicentre prospective clinical trial in NPC and other tumour types.

While it is widely conceived that baseline cfEBV DNA load may reflect occult metastasis burden in NPC[4,5], little information is

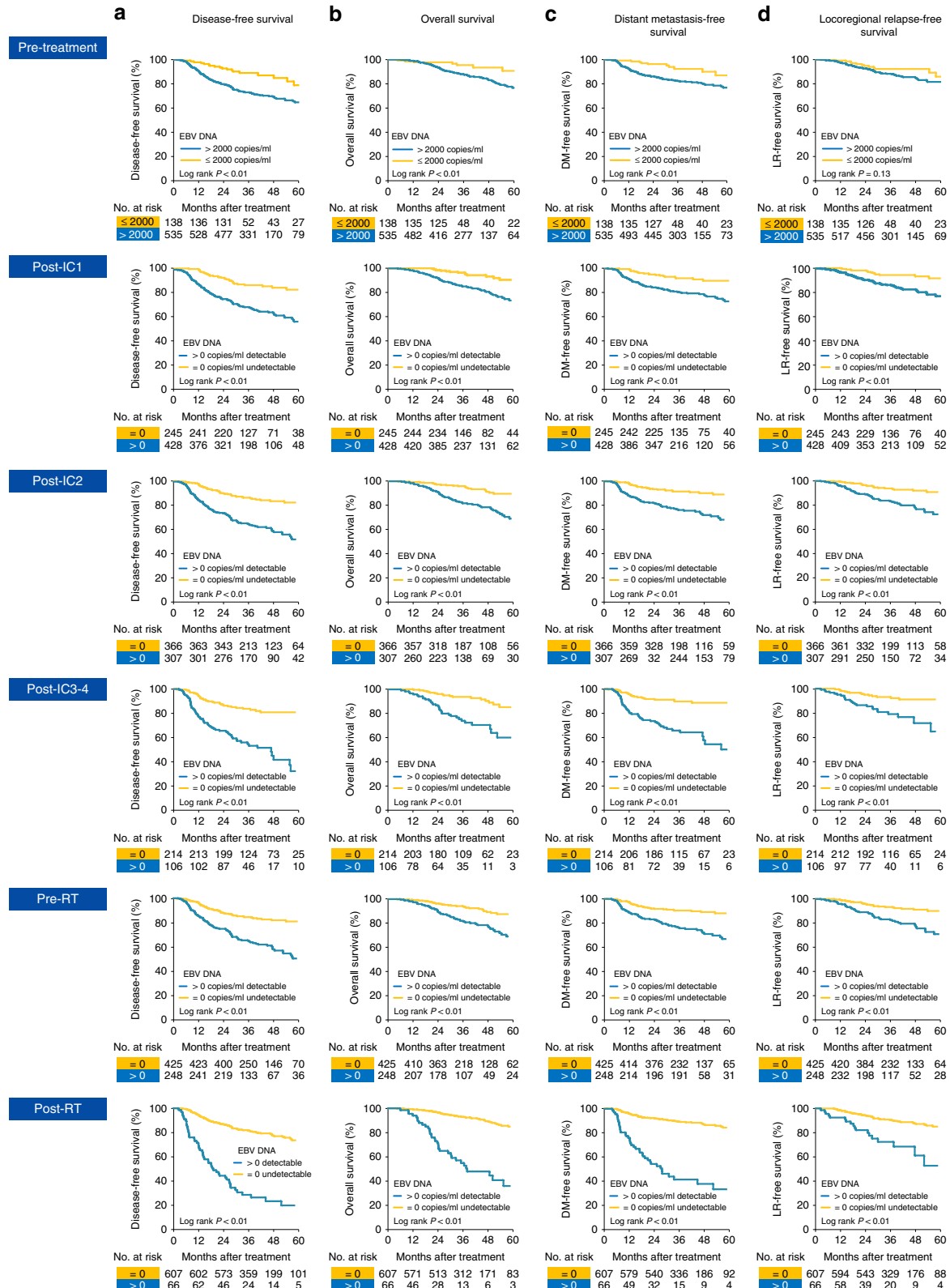

**Fig. 2** Kaplan–Meier plots of survival outcomes for subgroups achieving complete biological response at different time points. **a** Disease-free survival (DFS). **b** Overall survival (OS). **c** Distant metastasis-free survival (DMFS). **d** Locoregional relapse-free survival (LRFS). Time points included pretreatment, post-$IC_1$, post-$IC_2$, post-$IC_{3-4}$, pre-radiotherapy and post-concurrent radio-chemotherapy

available on clearance kinetics of the circulating tumour marker in response to treatment. Based on our detailed characterisation of the trajectory of biological responses to both IC and CCRT, we made the following key observations: foremost, a third of patients

achieved cBR after only one cycle of IC, and such extreme sensitivity to chemotherapy portends for a favourable prognosis. Second, the drop in cfEBV DNA was most acute after $IC_1$ across all groups, which would imply that the majority of tumour clones

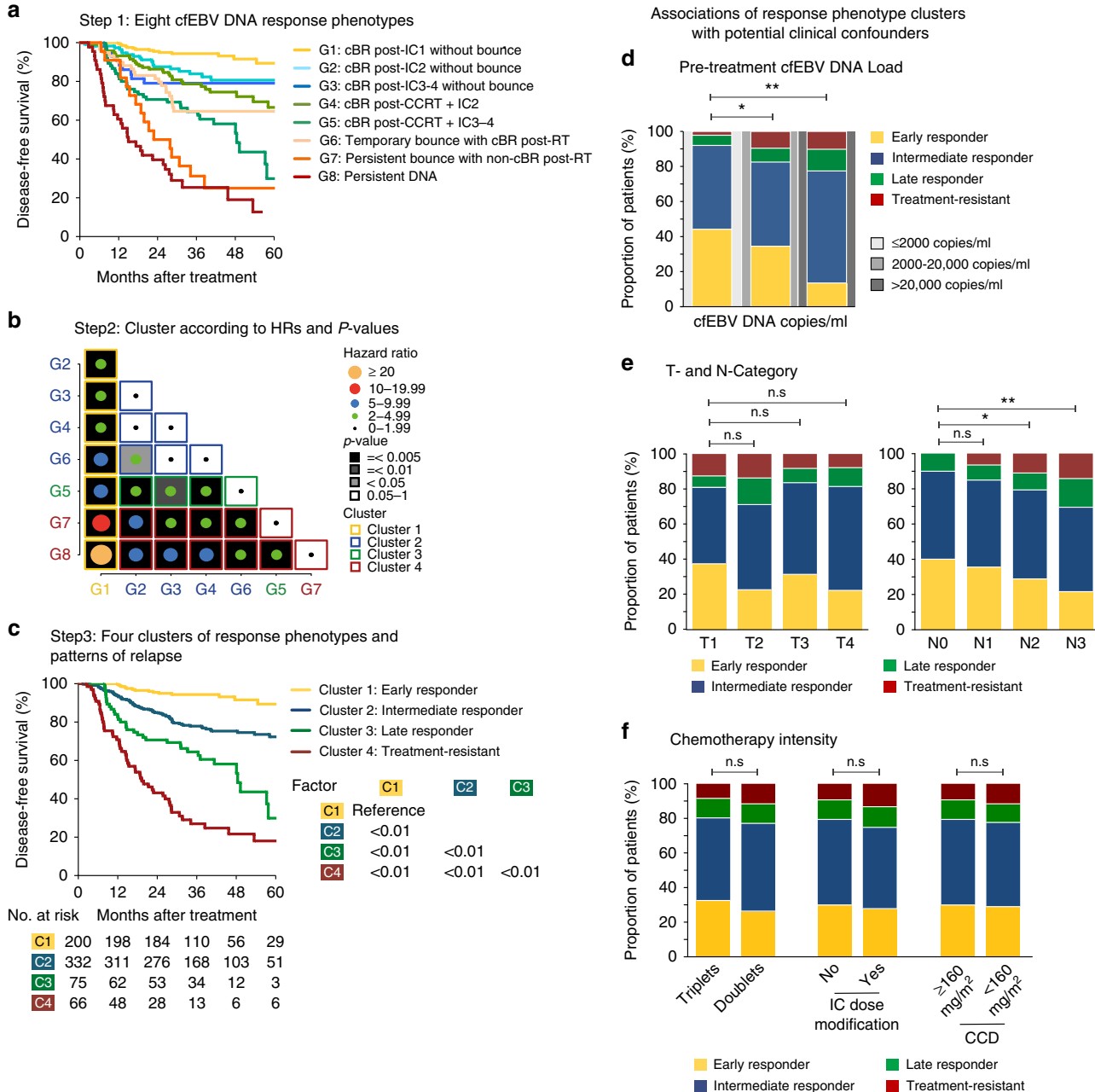

**Fig. 3** Survival clustering analyses of the different cell-free Epstein–Barr virus (cfEBV) DNA response phenotypes. **a** Step 1: Kaplan–Meier survival plot of disease-free survival (DFS) for the eight cfEBV DNA response phenotypes. **b** Step 2: Eight subgroups are then ordered in ascending numerical order and clustered by their intergroup hazard ratio (HR)$_{DFS}$. HRs were represented by the size of circle and categorised by steps of 5.0; $P$ values were represented in grey scale. We derived the following clusters: Cluster 1—G1, as G1 was significantly different from G2-8. Cluster 2—G2, G3, G4, G6, as G2-4 and G6 were not significantly different between them, except between G6 and G2. Cluster 3—G5, as G5 was significantly different from G1-4 and G8. Cluster 4—G7 and G8, as both groups were significantly different with G1-4, G6, except between G5 and G7. The clustering plot was done using ggplot2 package on *R*. **c** Step 3: Kaplan–Meier plot of DFS for the four phenotypic clusters. **d** Association of the phenotypes with pretreatment cfEBV DNA levels. **e** Association of the phenotypes with T and N categories, and **f** association of the phenotypes with chemotherapy intensity. cfEBV DNA levels were stratified based on previously reported cut-offs of ≤2000, >2000–20,000 and >20,000 copies/ml

harbour truncal genetic mutations that confer sensitivity to taxane, fluoropyrimidine, gemcitabine and/or platinum chemotherapy. However, this is followed by a delayed and slower rate of clearance over time, which supports our hypothesis that detectable biomarker post-IC$_1$ may reflect residual tumour clones that are resistant to therapies. Next, the differential responses to IC and CCRT among intermediate and late responders (e.g. non-cBR to IC but cBR to CCRT) suggest that tumour marker clearance during IC and CCRT are likely contributed by independent

mechanisms. Thus a suboptimal response to IC may not necessarily imply resistance to CCRT. In the same vein, we recorded occurrences of cfEBV DNA bounces during the IC and CCRT treatment phases, which had contrasting downstream implications; early bounces were mostly transient and likely to achieve cBR following CCRT, while delayed bounces were less likely to resolve and are indicative of early therapeutic resistance. Collectively, this and two previous studies[17,18] highlight the prognostic significance of cfEBV DNA clearance at various time points during treatment.

**Table 2 Pretreatment cfEBV DNA levels and absolute drop of cfEBV DNA during the course of treatment for the different subgroups**

| cfEBV DNA response phenotypes | Pretreatment cfEBV DNA (median, IQR) | Absolute cfEBV DNA drop during treatment ($\times 10^3$ copies/ml; median, IQR) | | | | |
|---|---|---|---|---|---|---|
| | | $IC_1$ phase | $IC_2$ phase | $IC_3$ phase | $IC_4$ phase | CCRT phase |
| *cfEBV DNA response groups* | | | | | | |
| G1 | 6.1 (1.2 to 20.5) | 6.1 (1.2 to 20.5) | 0.0 (0.0 to 0.0) | 0.0 (0.0 to 0.0) | 0.0 (0.0 to 0.0) | 0.0 (0.0 to 0.0) |
| G2 | 14.7 (2.0 to 98.5) | 10.9 (0.4 to 79.1) | 1.7 (0.6 to 7.0) | 0.0 (0.0 to 0.0) | 0.0 (0.0 to 0.0) | 0.0 (0.0 to 0.0) |
| G3 | 17.2 (2.3 to 62.0) | 13.4 (0.1 to 49.7) | 3.8 (1.0 to 12.9) | 1.5 (0.5 to 4.3) | 0.8 (0.2 to 1.7) | 0.0 (0.0 to 0.0) |
| G4 | 16.2 (5.2 to 68.5) | 7.7 (1.7 to 42.3) | 1.5 (−0.2 to 6.9) | — | — | 1.9 (0.5 to 7.5) |
| G5 | 41.1 (6.7 to 130.0) | 19.2 (2.0 to 77.9) | 4.0 (0.6 to 16.2) | 1.4 (0.4 to 6.0) | 1.0 (−2.3 to 1.1) | 1.9 (0.4 to 7.0) |
| G6 | 6.9 (1.9 to 25.0) | 6.6 (1.3 to 23.3) | 0.0 (−0.5 to 0.7) | 0.0 (−0.5 to 0.7) | 0.0 (−0.3 to 0.0) | 0.0 (0.0 to 0.6) |
| G7 | 18.1 (5.6 to 53.3) | 15.5 (3.4 to 38.8) | 0.3 (0.0 to 3.2) | 0.0 (0.0 to 0.0) | 0.0 (0.0 to 0.0) | −0.2 (−0.8 to 0.1) |
| G8 | 39.2 (11.2 to 201.0) | 9.5 (2.0 to 121.0) | 5.1 (1.2 to 27.4) | 3.0 (0.4 to 7.8) | 1.6 (0.2 to 4.1) | 0.9 (−2.6 to 24.5) |
| *cfEBV DNA phenotypic clusters* | | | | | | |
| Cluster 1 | 6.1 (1.2 to 20.5) | 6.1 (1.2 to 20.5) | 0.0 (0.0 to 0.0) | 0.0 (0.0 to 0.0) | 0.0 (0.0 to 0.0) | 0.0 (0.0 to 0.0) |
| Cluster 2 | 14.4 (3.0 to 55.6) | 9.0 (0.8 to 41.3) | 1.4 (0.1 to 5.8) | 0.0 (0.0 to 0.9) | 0.0 (0.0 to 0.9) | 0.0 (0.0 to 1.0) |
| Cluster 3 | 41.1 (6.7 to 130.0) | 19.2 (2.0 to 77.9) | 4.0 (0.6 to 16.2) | 1.4 (0.4 to 6.0) | 1.0 (−2.3 to 1.1) | 1.9 (0.4 to 7.0) |
| Cluster 4 | 29.8 (8.2 to 100.4) | 11.5 (2.8 to 72.3) | 3.0 (0.2 to 19.2) | 1.3 (0.0 to 7.1) | 1.3 (0.0 to 2.3) | −0.2 (−0.9 to 10.2) |

*cfEBV DNA* cell-free Epstein–Barr virus deoxyribonucleic acid, *CCRT* concurrent radio-chemotherapy, *IC* induction chemotherapy, *IQR* interquartile range
Note: G1, cBR post-IC1 without bounce; G2, cBR post-$IC_2$ without bounce; G3, cBR post-$IC_{3-4}$ without bounce; G4, cBR post-CCRT+$IC_2$; G5, cBR post-CCRT+$IC_{3-4}$; G6, temporary bounce with cBR post-CCRT; G7, persistent bounce with non-cBR post-CCRT; G8, persistent DNA. Cluster 1, early responders; Cluster 2, intermediate responders; Cluster 3, late responders; Cluster 4, treatment-resistance

Our observations also support a concept of utilising cfEBV DNA for individualisation of chemotherapy intensity in LA-NPC. Arguably, the conventional treatment strategy of LA-NPC entailed CCRT and AC[19,20]. Consequently, past and ongoing trials investigating the role of AC intensification have elected to stratify patients by cfEBV DNA post-CCRT. However, as observed in the NPC-0502 study, such an approach is inherently limited by the need to screen a large number of patients and the inability of patients to tolerate further chemotherapy in the adjuvant setting. With data now supporting the clinical efficacy of IC+CCRT in LA-NPC[12–15], it is intuitive to explore whether chemotherapy intensity can be modified accordingly at the earlier stages of treatment in the context of a prospective clinical trial (Fig. 5). Based on our phenotypic risk groupings, we proposed testing the omission of concurrent chemotherapy with radiotherapy in early responders (Arm I). In support, Xu et al. has previously demonstrated equipoise between sequential chemotherapy and radiotherapy and CCRT in low-risk LA-NPC (defined as non-T4N+ and/or N2–3)[21]. Next, we proposed investigating the integration of immune checkpoint inhibitor therapy with CCRT for the late responder subgroup (Arm III), to circumvent potential clonal chemo-resistance and accelerated tumour repopulation. This may be a trial-worthy approach given the reported synergy between concurrent immune checkpoint blockade and fractionated radiotherapy[22,23]. Ongoing clinical trials investigating such a therapeutic combination will provide preliminary data regarding its efficacy (NCT02952586, NCT02777385, NCT02586207, etc). Finally, we put forth an alternative approach of adjuvant metronomic chemotherapy or even immunotherapy in the treatment-resistant subgroup to improve tolerability and target residual resistant clones (Arm IV). A previous report by Twu et al. on the efficacy of metronomic oral tegafur–uracil in such high-risk patients[24] and recent data from the PACIFIC study in locally advanced lung cancer support this approach of systemic intensification[25].

Few limitations ought to be highlighted. Physician biases contributing to treatment heterogeneity were unavoidable. To best address this, we determined the association between known prognostic parameters and our derived phenotypes. Not surprisingly, early responders harboured lower pretreatment cfEBV DNA and nodal burden, but chemotherapy regime and drug intensity were comparable across the subgroups. In addition, while our response clusters were correlated with locoregional disease control, radiotherapy treatment parameters including time to initiation and treatment delays did not differ between them. Taken together, this would suggest that other mechanisms underpinning clonal sensitivity to chemotherapy and CCRT are likely implicated in the respective phenotypes. Admittedly, physician choice of treatment during the IC and CCRT phases may be biased by the cfEBV DNA result. However, time-dependent landmark and multivariable Cox regression analyses confirmed the prognostic significance of our response phenotypes after adjustment for clinical and treatment parameters. In the same vein, our institution protocol did not include clinical pathways targeted at non-cBR patients post-$IC_2$, and hence decision to proceed with further IC or CCRT was left to the discretion of the treating physician. It is uncertain whether this could have confounded our finding of inferior survival among the late responders. Next, as the frequency and timing of tumour marker assessment during CCRT were inconsistent, it precluded a detailed characterisation of biological responses during this phase of treatment. Lastly, the EBV DNA PCR-based assay is notoriously susceptible to significant inter-laboratory variations[26]. Nonetheless, our study cohort is comprised of patients who were treated at a single institution; we have previously reported robust quality control of our assay with minimal within-run (<10%) and between-day (<20%) variation[5].

In summary, we have identified four subsets of biological responders with disparate recurrence risks among LA-NPC cases treated with IC+CCRT. These response phenotypes may deserve different chemotherapy intensities for optimal distant metastasis control. We propose to incorporate these response phenotypes for stratification in a prospective clinical trial investigating the role of chemotherapy de-intensification and intensification in favourable and unfavourable LA-NPC, respectively.

## Methods

**Data extraction and patient selection.** Medical records of 10,126 non-metastatic NPC patients diagnosed between April 2009 and December 2015 were screened from an NPC-specific database within the Big-data intelligence framework at the Sun Yat-sen University Cancer Centre (SYSUCC). We retrieved data on 673 LA-NPC patients who received IC+CCRT and had a detectable pretreatment cfEBV DNA with longitudinal cfEBV DNA surveillance. The inclusion criteria included: (1) World Health Organisation type II or III NPC, positive for EBV viral capsid antigen (VCA/IgA) or EBV early antigen (IgA/EA); (2) LA-NPC (stage III–IVA)

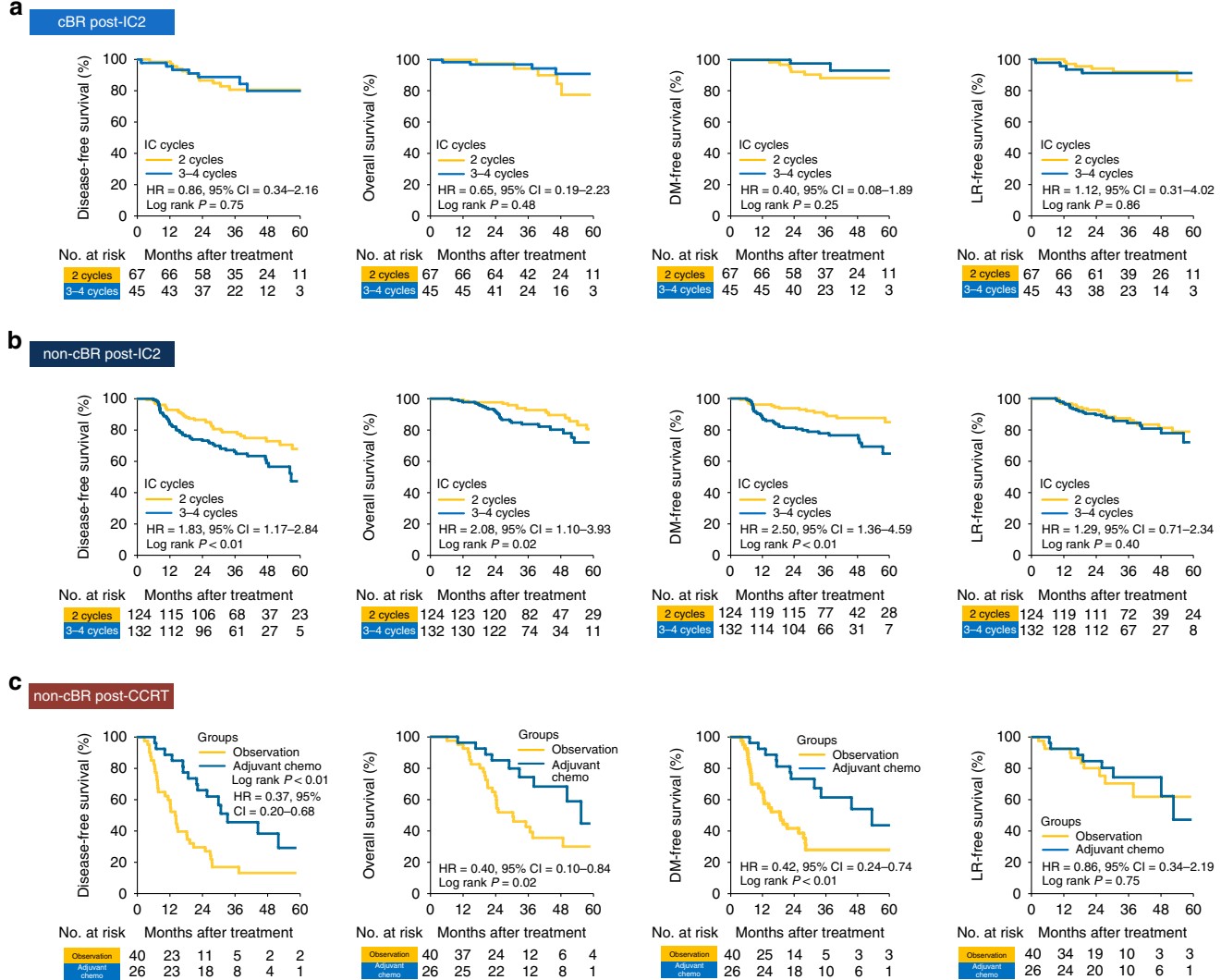

**Fig. 4** Survival outcomes with varying chemotherapy intensities for the respective response phenotypes. Disease-free (DFS), overall (OS), distant metastasis-free (DMFS), and locoregional relapse-free survival (LRFS) for **a** intermediate and late responders with complete biological response (cBR) post-IC$_2$, **b** intermediate and late responders without cBR post-IC$_2$ and **c** treatment-resistant subgroup

| Pretreatment cfEBV DNA | cfEBV DNA response | | Potential tumor biology | Real-time treatment adaptation | cfEBV DNA response | Real-time treatment adaptation |
|---|---|---|---|---|---|---|
| Detectable | cBR Post-IC1 | cBR Post-IC3 | | | cBR Post-RT | |
| Locoregionally advanced NPC  IC+CCRT (NCCN guideline) | Yes | Yes | Chemotherapy sensitive | Arm-I: Early responder (chemotherapy de-intensification) - Future RCT: IC+RT alone vs. IC+CCRT | Yes | Observation |
| | No | Yes | High volume of tumour burden | Arm-II: Intermediate responder - IC+CCRT | | |
| | Yes | No | Accelerated repopulation | Arm-III: Late responder (referred to CCRT and chemotherapy regimen alteration) - Future RCT: IC+CCRT-immune checkpoint-inhibitor(ICI) vs. IC+CCRT | No | Arm-IV: Treatment resistance (minimal residual disease) - Future RCT: IC+CCRT + AC (metronomic chemotherapy or ICI) vs. IC+CCRT |
| | No | No | Chemotherapy insensitive | | | |

**Fig. 5** Clinical trial utilising longitudinal cell-free Epstein–Barr virus (cfEBV) DNA response to individualise combination systemic treatment with radiotherapy. For early responders (Arm I), we proposed testing the omission of concurrent chemotherapy with radiotherapy. For intermediate responders (Arm II), we proposed maintaining chemotherapy intensity with concurrent radio-chemotherapy. For late responders (Arm III), alternative systemic combinations (e.g. immunotherapy) with radiotherapy rather than additional induction chemotherapy cycles ought to be considered. Lastly, for treatment-resistant subgroup (Arm IV), there may be a role for further adjuvant chemotherapy intensification with metronomic chemotherapy or immunotherapy

who were treated with IC plus CCRT±AC; (3) pretreatment EBV DNA >0 copies/ml; and (4) biomarker surveillance that involves cfEBV DNA quantification after every IC cycle, pre-CCRT and within 1 week at the end of CCRT (the collection schema is presented in Supplementary Fig. 2). To further account for treatment heterogeneity, we restricted the IC regimens to TPF, TP, PF and GP and

concurrent chemotherapy regimens to cisplatin and nedaplatin, which is consistent with the published literature[12,13,15]. The stepwise selection process and corresponding sample sizes are presented in Supplementary Fig. 1. All patients underwent required pretreatment evaluations and were restaged according to the Union for International Cancer Control/American Joint Committee on Cancer Eighth

edition stage classification system[27]. Details on the Big-data platform, patient selection process and diagnostic and staging work-up are presented in Supplementary Method. The study protocol was approved by the institutional ethics committee (No. YB2018-54). Informed consent was obtained from all patients. This study followed the REMARK guidelines (REporting recommendations for tumour MARKer prognostic studies).

**Treatment protocol and longitudinal cfEBV DNA surveillance**. The preferred first-line treatment for LA-NPC is IC+CCRT at SYSUCC, based on two positive randomised controlled phase 3 clinical trials favouring IC+CCRT in LA-NPC[12,13]. This is also in line with the updated 2018 National Comprehensive Cancer Network guideline recommendations (Category 2A)[27]. Radiotherapy and chemotherapy treatment protocols, including dose modifications, are described in Supplementary Method.

cfEBV DNA was measured at the following time points: 2-week before IC initiation (pretreatment), following every IC cycle (post-IC), during and within 1-week upon CCRT completion (post-CCRT). Details of cfEBV DNA quantification are described in Supplementary Method and Supplementary Fig. 2.

**Statistical analysis**. Primary endpoint was DFS. Secondary endpoints were OS, DMFS and LRFS. Survival curves were derived using the Kaplan–Meier method and compared by log-rank test[28]. Univariable and multivariable tests of association between the cBR phenotypes, clinical and treatment parameters and survival were performed using Cox regression. For the cBR phenotypes, we performed a supervised clustering based on relative intergroup $HR_{DFS}$ to reduce the number of subgroups. The proportional hazards assumption was verified using time-dependent Cox regression analysis[29]. Landmark analysis, which considers the time point when patients were placed into each classified group as the starting point for survival analysis (e.g. for early responders, the survival times were calculated starting from the initiation of the second IC cycle)[29], was also performed. The purpose of performing the landmark analysis was to address the possibility of physician bias in clinical decision-making that may be influenced by cfEBV DNA results during treatment. All analyses were performed with R version 3.4.4 (http://www.r-project.org) and SPSS version 23.0 (SPSS Inc., Chicago, IL, USA), with statistical significance set at two-sided $P < 0.05$. The clustering plot was done using the ggplot2 package on R. Detailed summary of the statistical considerations is presented in Supplementary Method.

## Data availability

The data regarding the baseline patient information, longitudinal EBV DNA copy number changes, survival outcomes and other detailed therapeutic information have been deposited in the Research Data Deposit public platform (www.researchdata.org.cn) under the accession code RDDA2019001004. All the other data supporting the findings of this study are available within the article and its supplementary information files and from the corresponding author upon reasonable request. A reporting summary for this article is available as a Supplementary Information file.

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

## Acknowledgements

The authors thank Professor Joseph Tien Seng Wee (from the Division of Radiation Oncology, National Cancer Centre Singapore) for his insightful comments on this study, members of the Chua and Soo Laboratories for their constructive comments on this manuscript and Yiducloud (Beijing) Technology Ltd. for the establishment of Big-data intelligence platform at Sun Yat-sen University Cancer Centre and their assistance during the data extraction process. This research supported by grants from the Pearl River Scholar Funded Scheme, the Special Support Program of Sun Yat-sen University Cancer Center (16zxtzlc06), the Health & Medical Collaborative Innovation Project of Guangzhou City, China (201604020003), the Natural Science Foundation of Guang Dong Province (No. 2017A030312003), Health & Medical Collaborative Innovation Project of Guangzhou City, China (201803040003), the Innovation Team Development Plan of the

Ministry of Education (No. IRT_17R110), the Overseas Expertise Introduction Project for Discipline Innovation (111 Project, B14035) and the National Natural Science Foundation of China (No. 81802707). M.L.K.C. is supported by the National Medical Research Council Singapore Clinician-Scientist Award - #NMRC/CSA/0027/2018 and the Duke-NUS Oncology Academic Program Proton Research Program. This research is also supported by the National Research Foundation Clinical Research Programme Grant (NRF-CRP17-2017-05).

## Author contributions

Conception and design: Y.S., M.L.K.C., J.L. Financial support: Y.S., M.L.K.C. Administrative support: Y.S., M.L.K.C., J.M. Provision of study materials or patients: Y.S., J.L. Collection and assembly of data: J.L., Y.C., G.Z., Z.Q., L.L., F.C., L.Z., X.H., R.L., S.X., Y.C. Data analysis and interpretation: J.L., Y.C., G.Z., Z.Q., K.R.L.T., H.W., M.L.K.C., Y.S. Manuscript writing: all authors. Final approval of manuscript: all authors.

## Additional information

**Competing interests:** M.L.K.C. reports personal fees from Astellas, personal fees from Janssen, grants and personal fees from Ferring, non-financial support from Astrazeneca, personal fees and non-financial support from Varian, grants from Sanofi Canada, grants from GenomeDx Biosciences and non-financial support from Medlever, outside the submitted work. The other authors declare no competing interests.

