## [Peer Review File · Nature Communications]

Reviewers' comments:

Reviewer #1 (Remarks to the Author)(NPC clinical studies):

In this study, the authors profile longitudinal cfEBV DNA responses to IC and CCRT in selected 673 LA-NPC (I assume EBV+) cases from a larger cohort of >10,000 biopsy proven, non-metastatic NPC derived from SYSUCC big data platform, and demonstrate temporal response to IC and CCRT is prognostic for DFS and may be utilized to risk stratify developing NPC clinical trials. The authors defined 8 phenotypes based on cfEBV DNA temporal response to IC+CCRT with differential DFS even when accounting for treatment and patient/tumor characteristics. These 8 phenotypes are further grouped into 4 clusters as early responders, intermediate responders, late responders and treatment resistant, which remained associated with DFS after adjusting for pre-treatment cfEBV load, chemotherapy intensity, T & N stage.

The work and paper is well-done and well-written. Statistical analyses are fine and appropriate. I would include the clinical outcomes data associated with OS, DMFS, LRFS in supplemental data to allow for more meaningful interpretation of presented data. The eligibility criteria to narrow the field from >10,000 to 673 should be further expanded on. What was the size and category of those ineligible based on search criteria used?

The findings are novel and of high interest to the NPC and larger head and neck cancer community. My primary question/concern is whether "responsiveness" is simply a marker of pre-treatment cfEBV DNA viral load? These absolute values are not available in the supplementary data. I assumed this data would be in sTable 2 or Figure 3D as authors mention in the text but I did not find it.

A few comments/questions:

- I assume all patients in this study completed the planned concurrent chemoRT course? Were there treatment interruptions?
- The authors mention the predominant pattern of failure among cluster 3 and 4 was distant metastases. Was there a difference in locoregional control among clusters?
- What was the absolute drop in cfEBV at each phase/phenotype? I think it is very important to show the median (range) pretreatment cfEBV among the 8 phenotypes and 4 clusters.
- Bounce phenomenon: this was a positive reading in the subsequent phase/measurement after cBR? Was the bounce phenomenon sustained throughout some or all of remaining IC cycles and/or CCRT? Please elaborate
- Any difference in outcomes among patients with bounce phenom after IC1-2?
- How many patients with cBRT after IC1-2 received cycles 3-4?
- How was persistent cfEBV defined in this study?
- Cluster 3/G5 is perhaps the most interesting subgroup/phenotype and appears to hint towards differences in treatment within this group. Out of curiosity: is data available to evaluate time/delays of therapy course, for example: if there were delays between induction and start of CCRT?
- Again please elaborate on screening criteria and how 673 were arrived at from initial database of >10,000.

Conclusion: Well-written and appropriate statistical analyses.

Novel finding of significant interest to the readership and cancer/cancer biology community at large. I would advocate publication in this journal. The aforementioned concerns should be addressed prior to publication.

Reviewer #2 (Remarks to the Author)(liquid biopsy, NPC):

The authors investigated liquid biopsy tracking on-treatment by longitudinally quantifying circulating Epstein-barr virus (EBV) DNA copy number in 673 nasopharynx cancer patients undergoing radical induction chemotherapy (IC) and chemo-radiotherapy (CRT). They present four prognostic phenotypes (early responders, intermediate responders, late responders, and treatment resistant) that were correlated with efficacy of chemotherapy intensity. The study is of high quality and the prognostic groups are interesting but exploratory and requires prospective validation. Overall, it represents an incremental advance over previous published work.

The results from a single institutional trial are not surprising. The persistence of a tumor marker after the 1st round and further on the 3rd round of treatment portends a worse prognosis. The potential value of these categories is in defining changes in therapy which are left to a new prospective trial.

1. The heterogeneity of treatment and lack of a defined protocol in this retrospective analysis can not control for treatment modification by the treating physicians. With increasing categories of resistance the number of patient decreases. This is clearly reflected in the extremely wide CI for DFS and OS. Such a wide gap would make defining treatment decision impossible without very large prospective data.

2. Rapid resolution of cDNA is a known prognostic factor in many tumor types including some very large studies previously published. This is the largest pool of data and the basis for the de-intensification consideration which appears to be an incremental advance over past work.

3. As pointed out, the frequency and timing of tumor marker assessment during CCRT were inconsistent, thus precluding a detailed characterisation of the responses during radiation therapy. This is an important limitation since patients go on to local radiation therapy as consolidation and patient response as well as any residual disease is also taken into consideration for radiation fields which can certainly affect outcomes.

4. The De-intensification approach can not be pursued outside of a very carefully controlled prospective clinical trial. There is a similar situation in head and neck cancers caused by the HPV virus. In HPV positive oropharyngeal cancers with better survival, these trials have proceeded cautiously, focusing on diminishing total radiation therapy for the most part, and are ongoing for years.

Authors' Responses

Reviewer #1 (NPC clinical studies):

Comment 1. In this study, the authors profile longitudinal cfEBV DNA responses to IC and CCRT in selected 673 LA-NPC (I assume EBV+) cases from a larger cohort of >10,000 biopsy proven, non-metastatic NPC derived from SYSUCC big data platform, and demonstrate that temporal response to IC and CCRT is prognostic for DFS and may be utilized to risk stratify developing NPC clinical trials. The authors defined 8 phenotypes based on cfEBV DNA temporal response to IC+CCRT with differential DFS even when accounting for treatment and patient/tumor characteristics. These 8 phenotypes are further grouped into 4 clusters as early responders, intermediate responders, late responders and treatment resistant, which remained associated with DFS after adjusting for pre-treatment cfEBV load, chemotherapy intensity, T & N stage.

The work and paper is well-done and well-written. Statistical analyses are fine and appropriate. I would include the clinical outcomes data associated with OS, DMFS, LRFS in supplemental data to allow for more meaningful interpretation of presented data. The eligibility criteria to narrow the field from >10,000 to 673 should be further expanded on. What was the size and category of those ineligible based on search criteria used?

Response: We thank the reviewer for the generous and positive comments on our work. To address the first point on providing granularity on data pertaining to other clinical end-points, we include the survival curves (OS, DMFS, and LRFS) for the eight biological response groups and four phenotypic clusters in **Figure below** (This is now added to the revised **Supplementary Figure 5**). The results of cox proportional hazards analyses of other clinical parameters associated with OS, DMFS, LRFS are included in the **Supplementary Table 4** of the manuscript. To summarise, we observe that our phenotypic clusters were associated with disparate survival

probabilities for OS, DMFS and LRFS, likewise to DFS. Of note, OS and DFS appeared to be more correlated with DMFS, although it is interesting to note that our phenotypic clusters were also linked to differences in local relapse. Cox regression analyses did not reveal any treatment-related confounders. We thank the reviewer for raising this point; we concur that the additional analyses provide a broader dimension to the interpretation of our data!

Supplementary Figure 5. Kaplan-Meier plots of overall (OS), distant metastasis-free (DMFS) and locoregional relapse-free survival (LRFS) for the eight cfEBV DNA response groups and four phenotypic clusters

Next, regarding the process of patient selection for this study, this was in fact shown in our initial submission (**Supplementary methods and Supplementary Figure 1**). To recap, the inclusion criteria included: (1) World Health Organization (WHO) type II or III NPC, positive for EBV viral capsid antigen (VCA/IgA) or

EBV early antigen (IgA/EA); (2) LA-NPC (stage III-IV_A) who were treated with IC plus CCRT ± AC; (3) pretreatment EBV DNA >0 copies/ml; and (4) biomarker surveillance that involves cfEBV DNA quantification after every IC cycle, pre-CCRT and within 1 week at the end of CCRT (the collection schema is presented in **Supplementary Figure 2**). To further account for treatment heterogeneity, we restricted the IC regimens to docetaxel/cisplatin/fluorouracil (TPF), docetaxel/cisplatin (TP), cisplatin/fluorouracil (PF), and gemcitabine/cisplatin (GP); and concurrent chemotherapy regimens to cisplatin and nedaplatin, which is consistent with the published literature. The stepwise patient selection process is presented in the **Figure** below and shown as **Supplementary Figure 1** in the manuscript. Nonetheless, to improve readability, we have moved the important points into the main text (please see **Page 16, paragraph 1**), and elaborated on the stepwise process in the revised manuscript.

Supplementary Figure 1. Flowchart showing the study design and patient selection process

Comment 2. The findings are novel and of high interest to the NPC and larger head and neck cancer community. My primary question/concern is whether "responsiveness" is simply a marker of pre-treatment cfEBV DNA viral load? These absolute values are not available in the supplementary data. I assumed this data would be in sTable 2 or Figure 3D as authors mention in the text but I did not find it.

Response: We thank the reviewer for the thoughtful comment, which we wholly agree with. In fact, we acknowledged this potential confounder, and performed analyses as shown in **Figure 3D panel 1**, which showed an association between

pre-treatment cfEBV DNA load and phenotypic clusters. Nonetheless, we agree that more granularity on this data will be useful for interpretation. In addition to the categorisation into < 2,000, 2,000-20,000 and > 20,000 copies/ml, we have included the median and interquartile range (IQR) of baseline cfEBV DNA levels for the eight response subgroups and four phenotypic clusters (**Table below**). Expectedly, median baseline cfEBV DNA for G1 (except G6 with temporary “bounce”) and Cluster 1 is the lowest in our cohort. Next, patients from G5 (cBR post CCRT after 3-4 cycles of IC) and G8 (persistent DNA), corresponding to Clusters 3 and 4 harboured the highest levels of cfEBV DNA. Overall, there appears to be a strong association between cfEBV DNA pre-treatment and clearance rate with treatment ($P < 0.01$, Kruskal-Wallis H test). Nonetheless, our four phenotype clusters remained significantly associated with prognoses, adjusted for pre-treatment cfEBV DNA load in the multivariable model.

In truth, a linkage between baseline tumour burden and rate of tumour clonal killing by chemotherapy and chemoradiotherapy is expected. However, we propose that other mechanisms underpinning clonal sensitivity to chemotherapy and chemoradiotherapy are likely at play, and collectively, these mechanisms define our biological phenotypes. We therefore design our prospective trial (as shown in **Figure 5**) based on these hypotheses, and future work will aim to define if indeed a mechanistic linkage exists between baseline tumour load and clonal sensitivity to treatment. We have included the table in our revised **Table 2**, and further highlighted these points in our revised **Discussion** section (please see **Page 14, Paragraph 2**).

Table 2. The distributions of pretreatment cfEBV DNA loads in eight cfEBV DNA response phenotypes and four cfEBV DNA response clusters

Phenotypes	Pre-treatment cfEBV DNA Load		P * -value
	Median	($\times 10^3$ copies/mL) Interquartile Range (IQR)	
cfEBV DNA Response Groups			< 0.01
G1	6.1	1.2–20.5	
G2	14.7	2.0–98.5	
G3	17.2	2.3–62.0	
G4	16.2	5.2–68.5	
G5	41.1	6.7–130.0	
G6	6.9	1.9–25.0	
G7	18.1	5.6–53.3	
G8	39.2	11.2–201.0	
cfEBV DNA Phenotypic Clusters			< 0.01
Cluster 1	6.1	1.2–20.5	
Cluster 2	14.4	3.0–55.6	
Cluster 3	41.1	6.7–130.0	
Cluster 4	29.8	8.2–100.4	

Note: G1, cBR post-IC1 without bounce; G2, cBR post-IC2 without bounce; G3, cBR post-IC3-4 without bounce; G4, cBR post-CCRT+IC2; G5, cBR post-CCRT+IC3-4; G6, temporary bounce with cBR post-CCRT; G7, persistent bounce with non-cBR post-CCRT; G8, persistent DNA.

Cluster 1, early responders; Cluster 2, intermediate responders; Cluster 3, late responders; Cluster 4, treatment-resistance.

* *P*-value was calculated by Kruskal-Wallis H test

Comment 3. I assume all patients in this study completed the planned concurrent chemoRT course? Were there treatment interruptions?

Response: We thank the reviewer for the query. Based on our elaborate patient selection process (please see **Supplementary Figure 1** above), it was required that all patients in this study completed radical intensity-modulated radiotherapy (IMRT).

Additionally, among the 673 patients enrolled, only 12/673 (1.8%) patients experienced radiotherapy interruption (please see **Table** below). We have added a mention of this point in our revised Results section and included the data as **Supplementary Table 3**.

Supplementary Table 3. List of patients with radiotherapy interruption

Treatment Interruption	No of patients (%)
Reason for Interruption	
Myelosuppression	7
Radiation-related adverse events*	3
Atrial fibrillation	1
Stomachache	1
Time of Interruption (Days)	
< 3	4
3-7	6
> 7	2
cfEBV DNA Response Phenotypes	
Cluster 1	4
Cluster 2	6
Cluster 3	1
Cluster 4	1

* included radiation-induced mucositis and dermatitis.

Comment 4. The authors mention the predominant pattern of failure among cluster 3 and 4 was distant metastases. Was there a difference in locoregional control among clusters?

Response: We thank the reviewer for the question. There is a trend toward favorable local control for cluster 3 compared with cluster 4, though no statistical significance was reached ($P = 0.14$, please see **Supplementary Figure 5** above), probably due to the limited sample sizes in these two groups. In this study, we observed that our phenotypic clusters were associated with disparate survival probabilities for OS, DMFS and LRFS, likewise to DFS. Of note, OS and DFS appeared to be more

correlated with DMFS, although it is interesting to note that our phenotypic clusters were also linked to differences in local relapse.

Comment 5. What was the absolute drop in cfEBV at each phase/phenotype? I think it is very important to show the median (range) pretreatment cfEBV among the 8 phenotypes and 4 clusters.

Response: We thank the reviewer for this comment, which we refer to our response to Comment #2. Additionally, we show the median drop in cfEBV DNA levels over the course of treatment (please see **Table** below). Interestingly, it highlights a few key observations; foremost, the drop in EBV DNA is most acute after IC₁ across all groups, which would imply that the majority of NPC tumour clones are sensitive to chemotherapy. Next, cfEBV DNA continues to drop with further treatment but the magnitude is substantially less compared to post-IC₁, which supports our hypothesis that detectable biomarker post-IC₁ likely reflects residual tumour clones that are resistant to therapies. We therefore designed our trial based on these scientific rationales. We have included these data in revised **Table 2** and discussed our key observations in the **Discussion** section (please see **Page 12, Paragraph 1**) of the revised manuscript. We thank the reviewer for this valuable comment, which has offered new insights on our data!

Table 2. The distributions of pretreatment cfEBV DNA loads and the absolute drop of cfEBV DNA during the course of treatment

cfEBV DNA Response Phenotypes	Pretreatment cfEBV DNA (Median, IQR)	Absolute cfEBV DNA drop during treatment (Median, IQR)				
		IC1 phase	IC2 phase	IC3 phase	IC4 phase	CCRT phase
cfEBV DNA Response Groups						
G1	6.1 (1.2–20.5)	6.1 (1.2–20.5)	0.0 (0.0–0.0)	0.0 (0.0–0.0)	0.0 (0.0–0.0)	0.0 (0.0–0.0)
G2	14.7 (2.0–98.5)	10.9 (0.4–79.1)	1.7 (0.6–7.0)	0.0 (0.0–0.0)	0.0 (0.0–0.0)	0.0 (0.0–0.0)
G3	17.2 (2.3–62.0)	13.4 (0.1–49.7)	3.8 (1.0–12.9)	1.5 (0.5–4.3)	0.8 (0.2–1.7)	0.0 (0.0–0.0)
G4	16.2 (5.2–68.5)	7.7 (1.7–42.3)	1.5 (-0.2–6.9)	--	--	1.9 (0.5–7.5)
G5	41.1 (6.7–130.0)	19.2 (2.0–77.9)	4.0 (0.6–16.2)	1.4 (0.4–6.0)	1.0 (-2.3–1.1)	1.9 (0.4–7.0)
G6	6.9 (1.9–25.0)	6.6 (1.3–23.3)	0.0 (-0.5–0.7)	0.0 (-0.5–0.7)	0.0 (-0.3–0.0)	0.0 (0.0–0.6)
G7	18.1 (5.6–53.3)	15.5 (3.4–38.8)	0.3 (0.0–3.2)	0.0 (0.0–0.0)	0.0 (0.0–0.0)	-0.2 (-0.8–0.1)
G8	39.2 (11.2–201.0)	9.5 (2.0–121.0)	5.1 (1.2–27.4)	3.0 (0.4–7.8)	1.6 (0.2–4.1)	0.9 (-2.6–24.5)
cfEBV DNA Phenotypic Clusters						
Cluster 1	6.1 (1.2–20.5)	6.1 (1.2–20.5)	0.0 (0.0–0.0)	0.0 (0.0–0.0)	0.0 (0.0–0.0)	0.0 (0.0–0.0)
Cluster 2	14.4 (3.0–55.6)	9.0 (0.8–41.3)	1.4 (0.1–5.8)	0.0 (0.0–0.9)	0.0 (0.0–0.9)	0.0 (0.0–1.0)
Cluster 3	41.1 (6.7–130.0)	19.2 (2.0–77.9)	4.0 (0.6–16.2)	1.4 (0.4–6.0)	1.0 (-2.3–1.1)	1.9 (0.4–7.0)
Cluster 4	29.8 (8.2–100.4)	11.5 (2.8–72.3)	3.0 (0.2–19.2)	1.3 (0.0–7.1)	1.3 (0.0–2.3)	-0.2 (-0.9–10.2)

Note: G1, cBR post-IC1 without bounce; G2, cBR post-IC2 without bounce; G3, cBR post-IC3-4 without bounce; G4, cBR post-CCRT+IC2; G5, cBR post-CCRT+IC3-4; G6, temporary bounce with cBR post-CCRT; G7, persistent bounce with non-cBR post-CCRT; G8, persistent DNA.

Cluster 1, early responders; Cluster 2, intermediate responders; Cluster 3, late responders; Cluster 4, treatment-resistance.

Comment 6. Bounce phenomenon: this was a positive reading in the subsequent phase/measurement after cBR? Was the bounce phenomenon sustained throughout some or all of remaining IC cycles and/or CCRT? Please elaborate.

& Comment 7. Any difference in outcomes among patients with bounce phenomenon after IC1-2?

Responses: We apologise for the ambiguity. To clarify, in this study we observed two bounce phenomena based on the time of onset: early bounce during IC phase (N = 39), and late bounce during CCRT phase (N = 42). Among 39 patients with early bounce, almost all bounces (N = 37) resolved post-CCRT; while among 42 patients with late bounce only 22 patients had a subsequent cBR post-CCRT (please see **main Figure 1**). We did not observe significant differences regarding survival outcomes between patients with early bounce and late bounce ($P_{DFS} = 0.15$); additionally, there were no differences regarding survival outcomes between bounce patients who had a cBR post-IC₁ versus post-IC₂ ($P_{DFS} = 0.83$). These findings suggested that early bounces were mostly transient and likely to achieve cBR following CCRT, while delayed bounces during CCRT were less likely to resolve and are indicative of early therapeutic resistance. Referring to the revised **main Table 2** (which shows the magnitude of cfEBV DNA clearance), we would like to refer to G7 (persistent bounce with non-cBR post-CCRT) to highlight a key observation; these individuals may not necessarily harbour high cfEBV DNA levels at baseline, and likewise to other subgroups, demonstrate an acute response to IC₁. However, these patients develop a detectable cfEBV DNA reading late into the treatment phase, which would indicate a repopulation of resistant tumour clones. Taken together, these findings may inform on the clonal trajectories in the recurrence process. To clarify this, we have refined the **main Figure 1** to better illustrate this idea.

Comment 8. How many patients with cBR after IC1-2 received cycles 3-4?

Response: We thank the review for this relevant question. The data were included in the old Supplementary Table 3 (**now Supplementary Table 5**) of our manuscript. 160 of 366 (43.7%) patients with cBR post-IC₂ received 3-4 cycles of IC, while 160 of 307 (52.1%) patients with non-cBR post-IC₂ received 3-4 cycles of IC ($P = 0.09$).

Supplementary Table 5. The number of IC cycles between patients with cBR and non-cBR post-IC₁₋₂

IC cycles	Post-IC ₁ cfEBVDNA		P -value*	Post-IC ₂ cfEBV DNA		P -value*
	cBR	Non-cBR		cBR	Non-cBR	
2 IC cycles	132 (53.9)	221 (51.6)		206 (56.3)	147 (47.9)	
3 IC cycles	88 (35.9)	180 (42.1)	0.10	134 (36.6)	135 (44.0)	0.09
4 IC cycles	25 (10.2)	27 (7.7)		26 (7.1)	25 (8.1)	

Abbreviations: cBR = complete biological response; IC = induction chemotherapy; cfEBV DNA = Epstein–Barr virus deoxyribonucleic acid.

* Two-sided *P*-values were calculated using the Chi-square test

Comment 9. How was persistent cfEBV defined in this study?

Response: Persistent cfEBV DNA is defined as detectable cfEBV DNA (>0 copies/mL) during the whole course of IC and CCRT. To clarify, we have added the definition in the **Results** section (please see **Page 6, Paragraph 2**) of the revised manuscript.

Comment 10. Cluster 3/G5 is perhaps the most interesting subgroup/phenotype and appears to hint towards differences in treatment within this group. Out of curiosity: is data available to evaluate time/delays of therapy course, for example: if there were delays between induction and start of CCRT?

Response: We concur with the reviewers that G5 is interesting, as conceptually it hints that additional induction chemotherapy is not effective in improving prognoses. To address the point of treatment confounders, we interrogated for the following factors: prolonged wait-time to radiotherapy, radiotherapy interruption, IC intensity, and concurrent chemotherapy dose. The wait-time of radiotherapy was calculated as the time intervals between the last cycle of IC and initiation of CCRT. The median wait-time of cluster 3 was 21 days (range, 19-31 days); 5/75 (6.7%) patients had wait-time longer than 33 days (delayed radiotherapy for more than one-week). Detailed information was presented in the table below.

List of wait-time of 75 patients in cfEBV DNA response cluster 3

Time of Intervals*	Delayed Time	No. of Patients (%)
21-23 days	On time	44 (58.7)
24-27 days	≤ 3 days	16 (21.3)
28-32 days	≤ 1 week	10 (13.3)
> 33 days	> 1week	5 (6.7)

* calculated by “the execution date of radiotherapy” minus “the execution date of the last cycle of IC”

The prolonged wait-time to radiotherapy, radiotherapy interruption, IC intensity, and concurrent chemotherapy dose of patients in different response clusters are comparable ($P > 0.05$, please see **Table** below). Therefore, we believe our characterisation of this subgroup to be biologically significant. We hypothesise that G5 likely harbour clonal chemo-resistance and early initiation of CCRT with novel agents (e.g., concurrent immunotherapy) may improve survival in this subgroup (**main Figure 5**). We have now included these data as **Supplementary Table 4** in the revised manuscript, and also added this point in the **Result** section (please see **Page 9, Paragraph 1**).

Supplementary Table 4. Comparison of the therapeutic confounders among patients in different cfEBV DNA response phenotypes

Therapeutic confounders	Cluster 1	Cluster 2	Cluster 3	Cluster 4	P-value*
Wait time					0.98
≤ 1 week	186 (93.0)	308 (92.8)	70 (93.3)	62 (93.9)	
> 1week	14 (7.0)	24 (7.2)	5 (6.7)	4 (6.1)	
Treatment Interruptions					0.97
No	196 (98.0)	326 (98.2)	74 (98.7)	65 (98.5)	
Yes	4 (2.0)	6 (1.8)	1 (1.3)	1 (1.5)	
IC regimens					0.28
Triplets	125 (62.5)	186 (56.0)	43 (57.3)	33 (50.0)	
Doublets	75 (37.5)	146 (44.0)	32 (43.7)	33 (50.0)	
CCD					0.79
≥ 160 mg/m ²	49 (24.5)	82 (24.7)	18 (24.0)	46 (30.0)	
<160 mg/m ²	151 (75.5)	250 (75.3)	57 (76.0)	20 (69.7)	

Abbreviations: CCD = cumulative concurrent chemotherapy dose; IC = induction chemotherapy

* Two-sided *P*-values were calculated using the Chi-square test

Comment 11. Again please elaborate on screening criteria and how 673 were arrived at from initial database of >10,000.

Response: We refer to our response to Comment #1. We appreciate the reviewer comment and have elaborated on a stepwise selection process in our revised manuscript (please see **Supplementary Figure 1**).

Comment 12. Conclusion: Well-written and appropriate statistical analyses. Novel findings of significant interest to the readership and cancer/cancer biology community at large. I would advocate publication in this journal. The aforementioned concerns should be addressed prior to publication.

Response: We thank the reviewer for the generous compliments. We agree that the comments raised interesting new insights on our data, and we have carefully addressed the aforementioned questions.

Reviewer #2 (liquid biopsy, NPC):

The authors investigated liquid biopsy tracking on-treatment by longitudinally quantifying circulating Epstein-barr virus (EBV) DNA copy number in 673 nasopharynx cancer patients undergoing radical induction chemotherapy (IC) and chemo-radiotherapy (CRT). They present four prognostic phenotypes (early responders, intermediate responders, late responders, and treatment resistant) that were correlated with efficacy of chemotherapy intensity. The study is of high quality and the prognostic groups are interesting, but exploratory, and requires prospective validation. Overall, it represents an incremental advance over previous published work.

Comment 1. The results from a single institutional trial are not surprising. The persistence of a tumor marker after the 1st round and further on the 3rd round of treatment portends a worse prognosis. The potential value of these categories is in defining changes in therapy, which are left to a new prospective trial.

& Comment 5. The De-intensification approach cannot be pursued outside of a very carefully controlled prospective clinical trial. There is a similar situation in head and neck cancers caused by the HPV virus. In HPV positive oropharyngeal cancers with better survival, these trials have proceeded cautiously, focusing on diminishing total radiation therapy for the most part, and are ongoing for years.

Response: We thank the reviewer for the kind comments. We concur on the importance of formally investigating the impact of our four prognostic phenotypes in treatment adaptation under the premise of a prospective clinical trial. Hence, we presented the concept of a potential trial based on our scientific observations in this paper in **Figure 5**. Specifically, we proposed to investigate the omission of concurrent chemotherapy after induction chemotherapy in early responders; explore the integration of immune checkpoint inhibitor (ICI) therapy with CCRT for late responders to circumvent potential clonal chemo-resistance and accelerated tumor repopulation; and test the efficacy of maintenance metronomic chemotherapy (capecitabine with/without ICI) for treatment-resistant patients.

In fact, the evidence in support for our de-intensification approach for cluster 1 was based on published data by our colleagues, who previously demonstrated equipoise between sequential chemotherapy and radiotherapy and concurrent CCRT in low-risk LA-NPC (defined as non-T4N+ and/or N2-3)¹. Moreover, in this study, early responders demonstrated promising survival outcomes, especially for DMFS,

and therefore omission of concurrent chemotherapy with RT may help reduce acute toxicity without compromising survival. Finally, as opposed to the de-intensification approaches that have been investigated in HPV+ oropharynx cancer, here, we are not proposing to reduce radiotherapy dose. We certainly share the concerns of the reviewer regarding the potential pitfalls of aggressive treatment de-intensification, and therefore we advocate for our prospective trial.

Comment 2. The heterogeneity of treatment and lack of a defined protocol in this retrospective analysis cannot control for treatment modification by the treating physicians. With increasing categories of resistance the number of patient decreases. This is clearly reflected in the extremely wide CI for DFS and OS. Such a wide gap would make defining treatment decision impossible without very large prospective data.

Response: Here, we fully concur with the reviewer, and are cognisant of this important limitation. To partly address this, we examined the association between our response phenotypes with the following confounders, including pretreatment cfEBV DNA load, T- and N-category, radiotherapy delay, radiotherapy interruption (suggested by the Reviewer #1), and chemotherapy intensity. Moreover, we did the time-dependent landmark analysis to help address physician bias in decision-making that may be influenced by cfEBV DNA results during treatment; multivariable Cox regression analysis was also adopted to confirm the prognostic significance of our response phenotypes.

Overall, as the reviewer pointed out, this is the first, and largest cohort of locally advanced NPC patients with dense longitudinal cfEBV DNA surveillance during radical treatment. Additionally, we applied a very careful and meticulous patient selection process that included controlling for IC regime and dose intensity, as well as radiotherapy dose. We feel that our data is practice changing, given that there is an emergence now among the community to rethink how to apply this liquid biopsy for risk stratification of LA-NPC patients². For future prospective data generation, we have initiated a prospective study – “Serial Epstein-Barr Virus DNA Surveillance in Non-metastatic Nasopharyngeal Carcinoma Patients during Radical Treatment” (Clinicaltrials.gov. NCT03855020) to collect more detailed information of serial on-treatment cfEBV DNA (especially cfEBV DNA during radiotherapy) in a larger cohort (N = 1,000). At the same time, we are planning to launch a cfEBV DNA-guided,

risk-adapted, phase II trial to explore the optimal treatment strategies for different response phenotypes.

Comment 3. Rapid resolution of ctDNA is a known prognostic factor in many tumor types including some very large studies previously published. This is the largest pool of data and the basis for the de-intensification consideration, which appears to be an incremental advance over past work.

Response: We thank the reviewer for the generous compliment.

Comment 4. As pointed out, the frequency and timing of tumor marker assessment during CCRT were inconsistent, thus precluding a detailed characterisation of the responses during radiation therapy. This is an important limitation since patients go on to local radiation therapy as consolidation and patient response as well as any residual disease is also taken into consideration for radiation fields, which can certainly affect outcomes.

Response: Here, we fully concur with the reviewer, and have in fact commented on this limitation in our Discussion. For the ongoing prospective observational study mentioned above, patients would have their blood taking for cfEBV DNA performed **weekly** during radiotherapy, which would provide insights on the clearance of this circulating biomarker to local treatment.

Next, we contest that radiation field variation could have influenced the survival among the delayed responders in this study. Foremost, limited number of patients in this study did replanning at the mid-course of radiotherapy 3/673 (0.4%), and only 11/673 (1.6%) patients received radiation boost. Next, our institutional radiotherapy guidelines recommend that delineation of gross disease is based on clinical and imaging examinations **prior to IC**, with the exception of tumour retraction from the nasal cavity. Nonetheless, there was a randomised trials that reported low rates of recurrence following volume reduction depending on response to IC³. Radiotherapy treatment delay was also infrequent in our cohort, and did not differ across clusters (**Supplementary Table 4** of the revised manuscript). We thank the reviewer for this comment, which prompted us to investigate deeply into the potential treatment confounders.

REFERENCE

1. Xu C, Sun R, Tang LL, Chen L, Li WF, Mao YP, *et al.* Role of sequential chemoradiotherapy in stage II and low-risk stage III-IV nasopharyngeal carcinoma in the era of intensity-modulated radiotherapy: A propensity score-matched analysis. *Oral oncology* 2018, **78**: 37-45.
2. Chua MLK. Circulating tumour DNA to personalize treatment in nasopharynx cancer – time to look “ahead”? *International journal of radiation oncology, biology, physics* 2019, **in press**.
3. Yang H, Chen X, Lin S, Rong J, Yang M, Wen Q, *et al.* Treatment outcomes after reduction of the target volume of intensity-modulated radiotherapy following induction chemotherapy in patients with locoregionally advanced nasopharyngeal carcinoma: A prospective, multi-center, randomized clinical trial. *Radiotherapy and oncology : journal of the European Society for Therapeutic Radiology and Oncology* 2018, **126**(1): 37-42.

REVIEWERS' COMMENTS:

Reviewer #2 (Remarks to the Author):

None

Reviewer #3 (Remarks to the Author):

This is a novel paper with solid statistical support for all the research claims. This paper is definitely acceptable for Nature communications.

Here are some specific comments which should be incorporated in the supplementary materials and methods to improve the manuscript.

1. nonhaematological toxicity
2. remove the word actuarial before survival
3. there should be a reference for log rank test
4. better to use the word univariate and multivariate instead of univariable and multivariable
5. clarify landmark analysis
6. In Supplementary Tables 4 onwards, many of the p-values are very large. There should be clear justifications mentioned in the text.

Authors' Responses

Reviewer #3:

This is a novel paper with solid statistical support for all the research claims. This paper is definitely acceptable for Nature communications.

Here are some specific comments, which should be incorporated in the supplementary materials and methods to improve the manuscript.

Comment 1. Nonhaematological toxicity.

Response: We thank the reviewer for the positive comments. While it may be useful information to report on non-haematological toxicities related to chemotherapy or radiotherapy, in truth, it somehow distracts the focus from the main messaging that is cfEBV DNA clearance and its association with survival and treatment efficacy. Moreover, we feel that the additional information does not supplement our analyses in this manuscript. We are currently planning a multi-arm phase II clinical trial based on our proposed design in **Figure 5**; toxicity outcome data will be collected in this upcoming prospective trial and will be reported with the main outcome data.

Comment 2. Remove the word actuarial before survival

Response: We thank the reviewer for this comment. We have checked the manuscript and confirm that there is no mention of actuarial throughout the manuscript.

Comment 3. There should be a reference for log rank test

Response: We thank the reviewer for this suggestion. We have now included a reference for log rank test in the revised manuscript. Please see **Reference No. 28**.

Comment 4. Better to use the word univariate and multivariate instead of univariable and multivariable

Response: We thank the reviewer for this comment. By statistical terminology, multivariate refers to an analysis whereby several distinct clinical outcomes are being simultaneously modeled, possibly as a function of one or more explanatory variables. Thus, it involves multifactorial modeling against multiple end-points of interest. Here,

we have simply performed a multivariable analysis, testing the association of EBV DNA response phenotypes and relevant prognostic factors in NPC separately for DFS, OS, LRFS and DMFS. Hence, the use of univariable and multivariable are the correct statistical terms here.

Comment 5. Clarify landmark analysis

Response: We thank the reviewer for this suggestion, and have included more details on landmark analysis in the Methods section of the revised manuscript (please see **Page 19, Paragraph 1**).

Comment 6. In Supplementary Tables 4 onwards, many of the p-values are very large. There should be clear justifications mentioned in the text.

Response: We thank the reviewer for this comment. The lack of association between treatment-related confounders and cfEBV DNA response phenotypes is further indicative that our characterisations of response phenotypes were biologically significant, and were not confounded by therapeutic parameters (including chemotherapy intensity, treatment interruption, and prolonged wait-time to radiotherapy). We have stated this finding in the Results section (please see **Page 10, Paragraph 1**), and also clarified this point in the Discussion section of the revised manuscript (please see **Page 15, Paragraph 2**).